# SEQR: Secure and Efficient QR-based LoRA Routing

## Abstract

Low-Rank Adaptation (LoRA) has become a standard technique for parameter-efficient fine-tuning of large language models, enabling large libraries of LoRAs, each for a specific task or domain. Efficiently selecting the correct LoRA adapter for a given input remains a challenge, particularly in secure environments where supervised training of routers may raise privacy concerns. Motivated by previous approaches, we formalize the goal of unsupervised LoRA routing in terms of activation norm maximization, providing a theoretical framework for analysis. We demonstrate the discriminative power of activation norms and introduce SEQR, an unsupervised LoRA routing algorithm designed to maximize efficiency while providing strict routing guarantees. SEQR provably identifies the norm-maximizing adapter with significantly greater efficiency, making it a highly scalable and effective solution for dynamic LoRA composition. We validate our results through experiments that demonstrate improved multi-task performance and efficiency. [1]

## 1 Introduction

Language model users can benefit from fine-tuning existing models on custom data, but may be constrained by security policies surrounding data access control or retention (Fleshman et al., 2024; Shi et al., 2025). Low-Rank Adaptation (LoRA) (Hu et al., 2022) is a popular parameter-efficient technique for fine-tuning these models. Widely-used software packages, such as `peft` (Mangrulkar et al., 2022), and model repositories, such as *huggingface* (Wolf et al., 2020), have contributed to the proliferation of LoRA-based experts fine-tuned for various tasks or data domains (Brüel-Gabrielsson et al., 2024; Huang et al., 2024). The broad deployment of language models has led to techniques for securing and controlling training data (Fleshman et al., 2024; Chowdhury et al., 2025; Shi et al., 2025). For example, ADAPTERSWAP leverages LoRA adapters to segment data into separate parameter groups, enabling user-based access control at the model level (Fleshman et al., 2024). The authorized LoRAs for a particular user can then be applied to the model at inference time, and adapters can be quickly retrained if training data is later removed to meet retention polices (Fleshman et al., 2024).

Naively applying all authorized LoRAs to a model can lead to parameter interference, significantly reducing the model performance (Wortsman et al., 2022; Chronopoulou et al., 2023; Ilharco et al., 2023; Fleshman et al., 2024). Numerous model merging strategies have been developed to address this challenge (Ortiz-Jimenez et al., 2023; Yadav et al., 2023; Tang et al., 2024; Yu et al., 2024; Stoica et al., 2025). Alternatively, LoRAs for the same model can be treated as a *mixture-of-experts* (Jacobs et al., 1991; Fedus et al., 2022) by learning to route inputs to a smaller set of appropriate adapters (Pfeiffer et al., 2021; Wang et al., 2022; Caccia et al., 2023; Ponti et al., 2023; Fleshman et al., 2024; Huang et al., 2024; Zadouri et al., 2024). Multi-LoRA frameworks have also been used for federated learning, where LoRA training dynamics suggest that the LoRA $A$ matrices learn global features which can be shared among the different adapters (Sun et al., 2024b; Guo et al., 2025).

Supervised training of a router using data across protected silos is not an option in strict data security scenarios, as adversarial techniques exist for leaking information related to the data (Shokri et al., 2017; Carlini et al., 2022; Yao, 2024; Zhou et al., 2025). Recent approaches perform LoRA routing in an unsupervised manner by selecting adapters for a given input without any router training or cross-silo data requirements (Ostapenko et al., 2024; Fleshman & Van Durme, 2025a;b). In this work,

---

[1]Code available at https://anonymous.4open.science/r/SEQR

Figure 1: Secure and Efficient QR (SEQR) routing: Rank-$r$ LoRAs are trained on multiple datasets using a shared $A$ matrix frozen at initialization. Each $B_i$ is stored in terms of its QR decomposition. During inference, input vectors are routed efficiently using the smaller $r \times r$ matrices.

we formalize the goal of these techniques and analyze their routing procedures. We introduce a new method, SEQR (Figure 1), which is more efficient than previous approaches while providing strict routing guarantees. Specifically we:

- Formalize unsupervised LoRA routing as activation norm-maximization;
- Provide theoretical results for current approaches under this framework;
- Introduce a more efficient routing scheme, SEQR, which provably selects the norm-maximizing adapter; and
- Perform empirical experiments demonstrating the benefits of our approach.

## 2 BACKGROUND AND RELATED WORK

### 2.1 LORA

LoRA updates the pretrained layer weights $W_0 \in \mathbb{R}^{m \times n}$ by freezing the existing weights and injecting two low-rank matrices of learnable parameters $A \in \mathbb{R}^{r \times n}$ and $B \in \mathbb{R}^{m \times r}$ such that the new weights are $W = W_0 + BA$, with a small rank $r \ll \min(m, n)$ that considerably reduces the number of trainable parameters (Hu et al., 2022). For an input vector $\mathbf{x} \in \mathbb{R}^n$, the output $\mathbf{y} \in \mathbb{R}^m$ can be computed directly with the new weights as $\mathbf{y} = W\mathbf{x}$ or separately as $\mathbf{y} = W_0\mathbf{x} + BA\mathbf{x}$. LoRA fine-tuning has been widely adopted, and the success has led to many extensions to the original idea (Zhang et al., 2023b; Albert et al., 2024; Buehler & Buehler, 2024; Kopiczko et al., 2024; Li et al., 2024; Liu et al., 2024a; Bini et al., 2025). LoRA merging or routing is necessary in the case where many LoRAs are trained on different groups of data, resulting in a set of available LoRAs $\mathcal{A} = \{B_1A_1, B_2A_2, \ldots, B_NA_N\}$ for each adapted layer of the model. Various techniques for merging LoRAs into a single update have been developed, but merging suffers from parameter interference when the number of adapters is large (Ortiz-Jimenez et al., 2023; Yadav et al., 2023; Tang et al., 2024; Yu et al., 2024; Stoica et al., 2025). The goal of unsupervised LoRA routing is to choose the subset of adapters best suited for each vector in a sequence, without explicitly training a router (Ostapenko et al., 2024; Fleshman & Van Durme, 2025b;a). These approaches alleviate parameter interference while providing additional security guarantees (Fleshman & Van Durme, 2025b).

### 2.2 PRIVACY & SECURITY

Organizations may have various security or privacy concerns depending on the data used for training individual LoRAs. Training with differential privacy (DP) provides probabilistic guarantees that an adversary can not infer if particular examples were in the training data (Dwork & Roth, 2014; Abadi et al., 2016). DP can be used to protect user privacy in cases where adversaries may have access to the LoRA weights (Shi et al., 2025). Stricter security requirements incorporate data access control, completely preventing user access to LoRA weights trained on data the user is unauthorized

to view (Fleshman et al., 2024). In these cases, training a router to distinguish between LoRAs would introduce security concerns, as adversaries with access to the router could potentially leak information from the LoRAs themselves (Shokri et al., 2017; Carlini et al., 2022; Yao, 2024; Zhou et al., 2025). We focus on the strict security case, where unsupervised routing approaches are needed.

## 2.3 ACTIVATION NORMS

Unsupervised LoRA routing can be framed as an *in-distribution* (ID) detection problem, where inputs are routed to the adapters trained on data similar to the queries. Prior work has shown that the norm of the activation vector produced by model layers can effectively distinguish between in- and out-of distribution (OOD) data (Park et al., 2023; Liu et al., 2024b; Shin & Chung, 2024; Sun et al., 2024a; Wan et al., 2024). ID data tends to produce large activation spikes in neural networks, including in large language models (Sun et al., 2024a). Park et al. (2023) analyze this phenomenon and find that the activation norm distinguishes OOD and ID similar to a classifier confidence score. These findings justify trying to route to LoRAs which maximize the norm of adapter activations $||BA\mathbf{x}||$.

## 2.4 ARROW ROUTING

Ostapenko et al. (2024) use the singular value decomposition (SVD) to convert each LoRA adapter $B_iA_i \in \mathcal{A}$ into a product of three matrices with an equivalent product:

$$B_iA_i = U_iS_iV_i^T, \tag{1}$$

where $U_i \in \mathbb{R}^{m \times r}$ is the orthonormal matrix of left singular vectors, $S_i \in \mathbb{R}^{r \times r}$ is the diagonal matrix of singular values, and $V_i \in \mathbb{R}^{n \times r}$ is the orthonormal matrix of right singular vectors. ARROW routing leverages the fact that the right singular vector $\mathbf{v}_i$ associated with the largest singular value corresponds to the direction capturing the most variation in the space of input vectors $\mathbf{x}$ (Ostapenko et al., 2024). This *arrow vector* $\mathbf{v}_i$ satisfies $\mathbf{v}_i = \max_{\mathbf{x},||\mathbf{x}||_2=1}||B_iA_i\mathbf{x}||_2$, meaning it maximizes the norm of the corresponding adapter activations among unit-length input vectors. We use norm-maximization as the explicit goal in this work, allowing for analysis of these approaches. Ostapenko et al. (2024) use the set of arrows as prototypes for each of the adapters in $\mathcal{A}$, assigning the most weight to the adapter corresponding to the arrow satisfying $\text{argmax}_i|\mathbf{v}_i^T\mathbf{x}|$. The use of vector prototypes makes ARROW routing especially efficient, requiring a simple dot product per adapter: $\mathcal{O}(Nn)$ for $N$ adapters with input dimension $n$. ARROW routing performs reasonably well, and the authors empirically show that the ID adapter tends to produce higher ARROW scores (Ostapenko et al., 2024).

## 2.5 SPECTRAL ROUTING AND LAG

SPECTR builds on ARROW by using all right singular vectors to make routing decisions (Fleshman & Van Durme, 2025b). Equation 1 is used by SPECTR to convert each adapter into two new matrices:

$$\hat{B}_i = U_i \tag{2}$$

$$\hat{A}_i = S_iV_i^T, \tag{3}$$

such that $\hat{B}_i\hat{A}_i = B_iA_i$ with $\hat{A}_i$ now containing the orthogonal directions of maximum variation scaled by the singular values. SPECTR generalizes the ARROW scoring method by assigning the most weight to the adapter satisfying $\text{argmax}_i||\hat{A}_i\mathbf{x}||_2$. Computing the SPECTR routing scores is less efficient than ARROW: $\mathcal{O}(Nrn)$, but SPECTR outperforms ARROW in routing accuracy and downstream task performance (Fleshman & Van Durme, 2025b).

LoRA-Augmented Generation (LAG) combines the efficiency of ARROW routing with the improved performance of SPECTR by using a two-stage approach (Fleshman & Van Durme, 2025a). First, LAG performs top-$k$ filtering using ARROW to reduce the final routing decision to $k \ll N$ adapters. LAG then uses SPECTR to route to the top adapter in the filtered set. Routing complexity is reduced to $\mathcal{O}(Nn + krn)$ while still outperforming ARROW (Fleshman & Van Durme, 2025a).

## 2.6 SHARED A

While ARROW, SPECTR, and LAG use traditional LoRA fine-tuning, recent work explores a special case of LoRA where the $A$ matrix is frozen at initialization or shared among several LoRAs in a

federated setting, resulting in similar or improved performance with reduced storage costs (Zhang et al., 2023a; Sun et al., 2024b; Zhu et al., 2024; Guo et al., 2025). Zhu et al. (2024) provides a theoretical analysis showing that the LoRA updates are dominated by the B matrix during fine-tuning, and that a LoRA with a frozen random $A$ matrix should perform similarly to one that is fully trained. The asymmetry in training dynamics lends itself to using a global $A$ matrix and unique $B$ matrices in multi-LoRA scenarios (Sun et al., 2024b; Guo et al., 2025). We explore this direction in our work, and show that a shared $A$ matrix allows for more efficient unsupervised LoRA routing techniques.

# 3 THEORETICAL RESULTS AND SEQR

**Problem Statement** We formalize the goal of unsupervised LoRA routing to provide a framework for theoretical analysis. Given the success of using activations for ID/OOD detection and the similar motivation of current unsupervised routing approaches, we propose the following problem:

---
**Problem**

**LoRA Activation Norm-Maximization.** Given a library of LoRA adapters, $\mathcal{A} = \{B_1 A_1, B_2 A_2, \ldots, B_N A_N\}$ and an input vector $\mathbf{x}$, efficiently find $\mathrm{argmax}_i ||B_i A_i \mathbf{x}||_2$.

---

We add "efficiently" to the problem statement as an algorithm that simply computes all activation norms directly would be $\mathcal{O}(Nr(m+n))$, far worse than current routing approaches. We will demonstrate the discriminative power of LoRA activation norms in Section 4.3.

## 3.1 ARROW IS NOT NORM-MAXIMIZING

Our first result shows that ARROW is not guaranteed to find the norm-maximizing adapter.

**Theorem 3.1.** *There exists a set of LoRA adapters* $\{B_1 A_1, B_2 A_2, \ldots, B_N A_N\}$ *with corresponding arrow vectors* $\{v_1, v_2, \ldots, v_N\}$ *and* $\mathbf{x} \in \mathbb{R}^n$ *where* $\mathrm{argmax}_i |\mathbf{v}_i^T \mathbf{x}| \neq \mathrm{argmax}_i ||B_i A_i \mathbf{x}||_2$.

We provide the proof by construction in Appendix A and confirm with experiments in Section 4.4. The main observation from the proof is that alignment with the top singular vector is not enough to guarantee the adapter will have the largest norm, as misalignment can be overcome with larger singular values. Routing with LAG inherits the lack of guarantee from ARROW, but the top-$k$ selection improves the chances of including the norm-maximizing adapter in the set used for SPECTR selection.

## 3.2 SPECTR IS NORM-MAXIMIZING

Our next results show that SPECTR scores are equivalent to the activation norms, and therefore SPECTR is norm-maximizing. The proof for Theorem 3.2 is provided in Appendix B.

**Theorem 3.2.** *Let* $B \in \mathbb{R}^{m \times r}$ *and* $A \in \mathbb{R}^{r \times n}$ *be LoRA matrices with* $\hat{A}$ *derived from* $BA$ *using Equations 1 and 3, then* $\forall \mathbf{x} \in \mathbb{R}^n$, $||\hat{A}\mathbf{x}||_2 = ||BA\mathbf{x}||_2$.

**Corollary 3.2.1.** *Let* $\{B_1 A_1, B_2 A_2, \ldots, B_N A_N\}$ *be a set of LoRA adapters converted with Equations 1-3 to the set* $\{\hat{B}_1 \hat{A}_1, \hat{B}_2 \hat{A}_2, \ldots, \hat{B}_N \hat{A}_N\}$, *then* $\mathrm{argmax}_i ||\hat{A}_i \mathbf{x}||_2 = \mathrm{argmax}_i ||B_i A_i \mathbf{x}||_2$.

These results show that SPECTR provides optimal routing under the stated goal. We are interested in new approaches providing the same guarantee but with improved efficiency.

## 3.3 SECURE AND EFFICIENT QR (SEQR) ROUTING

Now we explore the special case of our problem statement where all adapters in $\mathcal{A}$ share the same matrix $A$. This matrix is randomly initialized and kept frozen to ensure the same data security provided by other unsupervised routing approaches. For an input $\mathbf{x}$, we compute $\mathbf{z} = A\mathbf{x}$ as an intermediate step. Routing is then required for the set of $B$ matrices. Directly computing the norm for all would require $\mathcal{O}(Nmr)$, which is already equivalent to SPECTR for $m = n$. We can improve further by doing a one-time preprocessing step similar to the SVD in ARROW and SPECTR. We precompute the reduced QR decomposition of each $B_i$:

$$B_i = Q_i R_i, \tag{4}$$

where $Q_i \in \mathbb{R}^{m \times r}$ is an orthogonal matrix and $R_i \in \mathbb{R}^{r \times r}$ is upper triangular. Similar to SPECTR, we can throw away the original $B_i$ and store the adapter in this new form. The vector $\mathbf{z}$ is then routed to the adapter satisfying $\text{argmax}_i ||R_i \mathbf{z}||_2$.[2] The routing complexity is only $\mathcal{O}(Nr^2)$, which is far better than SPECTR and is even more efficient than ARROW routing in the typical LoRA scenario where $r \ll n$. We restate the complete SEQR routing algorithm in Appendix F. Like SPECTR, we show SEQR scores are equivalent to the activation norm for each adapter. Therefore, SEQR always selects the norm-maximizing adapter. The proof for Theorem 3.3 is provided in Appendix C.

**Theorem 3.3.** *Let $B \in \mathbb{R}^{m \times r}$ and $A \in \mathbb{R}^{r \times n}$ be LoRA matrices such that $B = QR$ from Equation 4, then $\forall \mathbf{x} \in \mathbb{R}^n$, $||RA\mathbf{x}||_2 = ||BA\mathbf{x}||_2$.*

**Corollary 3.3.1.** *Let $\{B_1, B_2, \ldots, B_N\}$ be a set of LoRA adapters with a shared $A$ matrix and $\{Q_1 R_1, Q_2 R_2, \ldots, Q_N R_N\}$ from Equation 4, then $\text{argmax}_i ||R_i A\mathbf{x}||_2 = \text{argmax}_i ||B_i A_i \mathbf{x}||_2$.*

### 3.4 ROUTING COMPLEXITY

We revisit the routing complexities of ARROW routing, SPECTR, LAG, and SEQR using dimensions reported in the LAG experiments for added context (Fleshman & Van Durme, 2025a). Table 1 includes the FLOPs used for routing by each method in this example, including the naive approach of computing the norm directly for each adapter. SEQR is two orders of magnitude more efficient than any other approach. SEQR also decreases storage costs by offsetting the storage of each $R_i$ by sharing $A$ across the library. ARROW can also take advantage of improved storage when using a shared $A$ matrix, but arrow vectors require more space than the $R_i$ matrices when $n > r^2$.

Table 1: Routing complexity and example FLOPs for each method assuming $N = 1000$ adapters, $n = m = 4096$ hidden dimension, $k = 20$ LAG filtering, and $r = 8$ rank adapters.

|  | Naive | SPECTR | LAG | ARROW | SEQR |
|---|---|---|---|---|---|
| **FLOPs** | 66M | 33M | 5M | 4M | **64K** |
| **Complexity** | $\mathcal{O}(Nr(m+n))$ | $\mathcal{O}(Nrn)$ | $\mathcal{O}(Nn+krn)$ | $\mathcal{O}(Nn)$ | $\mathcal{O}(Nr^2)$ |

## 4 EXPERIMENTS

We conduct experiments to validate our theoretical results and to test whether SEQR provides similar or better performance over less efficient alternatives. First, we confirm prior work showing that using a fixed $A$ matrix in LoRA works as well as learning $A$ individually. We analyze the differences in activation norms between these two settings and introduce a calibration step to ensure norms between adapters are on the same scale. We measure the ability of each approach to select the norm-maximizing adapter and the resulting multi-task performance and efficiency.

### 4.1 MODELS AND DATA

We first replicate the experiments of Fleshman & Van Durme (2025b) using the Llama-3.2-3B-Instruct model (Grattafiori et al., 2024). We train LoRAs for a variety of tasks: agnews[3], cola (Warstadt et al., 2019), dbpedia (Auer et al., 2007), hellaswag (Zellers et al., 2019), mnli (Williams et al., 2018), mrpc (Dolan & Brockett, 2005), qnli (Rajpurkar et al., 2016), qqp,[4] rte (Wang et al., 2018), and sst2 (Socher et al., 2013). Similar to Ostapenko et al. (2024), we subsample the datasets for computational feasibility. Using different random seeds, we produce three sets of LoRAs per dataset and category (shared vs. unique $A$ matrix), each trained on 1000 samples from the corresponding dataset. Learning rates were optimized per dataset and category but shared across random seeds. All evaluations are performed using a held-out set of 1000 examples from each dataset. LoRA $A$ matrices are initialized from $\mathcal{N}(0, 1/r^2)$ and frozen in the shared setting. The $B$ matrices are initialized with 0s and trained in both cases (Hu et al., 2022). The complete adapter training details are included in Appendix D. Finally, we demonstrate SEQR across different model sizes and families using the Qwen 1.5B, Llama 3B, Qwen 7B, and Llama 8B instruct models (Grattafiori et al., 2024; Qwen et al., 2025).

---

[2]We z-score these raw scores based on our findings in Section 4.3.

[3]http://groups.di.unipi.it/~gulli/AG_corpus_of_news_articles.html

[4]https://quoradata.quora.com/First-Quora-Dataset-Release-Question-Pairs

Table 2: Accuracy of Llama-3B LoRAs using a unique or fixed $A$ matrix shared across datasets.

| | agnews | cola | dbped | hswag | mnli | mrpc | qnli | qqp | rte | sst2 | AVG |
|---|---|---|---|---|---|---|---|---|---|---|---|
| Unique | 90.4 | 78.8 | 98.7 | 83.6 | 86.1 | 84.7 | 84.9 | 86.5 | 88.2 | 92.4 | 87.4 |
| Shared | 90.0 | 78.9 | 99.0 | 81.5 | 85.7 | 85.0 | 85.5 | 86.3 | 87.9 | 92.8 | 87.3 |

## 4.2 UNIQUE VS. SHARED

Before measuring routing performance, we ensure that using frozen $A$ matrices results in similar LoRA performance. Table 2 shows the accuracy of each adapter on its corresponding test set, averaged across three initializations. Accuracy is within 1% between the two categories in most cases, with the largest deviation being a 2% difference on hellaswag when using the frozen $A$ matrices. Overall, the average performance is nearly identical, a finding consistent with prior work showing similar performance with a frozen $A$ (Zhang et al., 2023a; Sun et al., 2024b; Zhu et al., 2024).

## 4.3 ACTIVATION NORMS

Activation norms of a given adapter can be informative for distinguishing ID from OOD data. However, to ensure bias-free routing, these norms must be comparable across adapters. For instance, the agnews adapter may produce lower norms than the cola adapter regardless of the dataset, even if it generates higher norms on agnews data specifically. In such cases, the routing procedure would be biased toward selecting the cola adapter. We explore and mitigate this potential bias in norms.

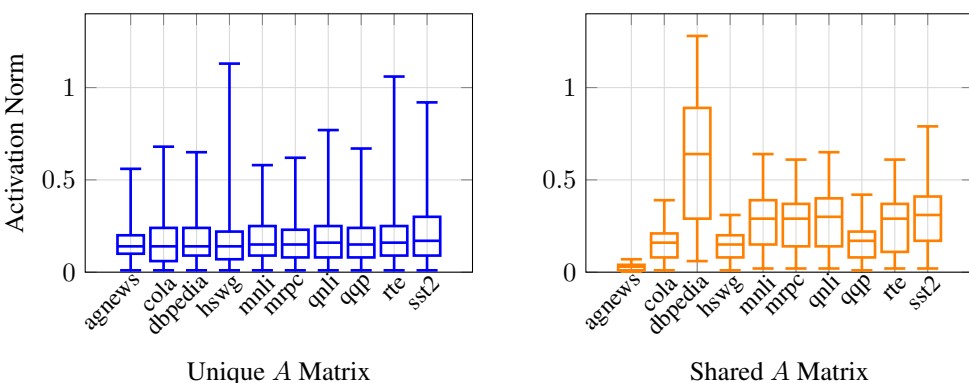

Figure 2: Distribution of average activation norms for each dataset when using LoRA adapters with unique $A$ matrices or a fixed $A$ matrix shared across adapters.

We gather the average activation norms across model layers for each adapter in Figure 2. We find that the activation norms are very consistent across adapters when using LoRAs trained with unique $A$ matrices. However, when the adapters share a frozen $A$ matrix across datasets, the variance in activation norms is considerable. To address the issue, we introduce an offline calibration step for the norm-based approaches. We compute the mean, $\mu_i$, and standard deviation, $\sigma_i$, of the activation norm for each adapter using its associated training data. The scores for SPECTR become $s_i = (||\hat{A}_i \mathbf{x}||_2 - \mu_i)/\sigma_i$ and similarly for SEQR $s_i = (||R_i A \mathbf{x}||_2 - \mu_i)/\sigma_i$, which are the z-scores of the original raw scores to ensure all adapters are on the same scale. These scores are used in the final algorithm (Appendix F). ARROW scores remain the same, as $\mathbf{v}_i$ is unit-length by construction.

We visualize the impact of z-scoring by measuring the average activation norm for each adapter, on each dataset, before and after normalizing the scores (Figure 3). We see that the norms for adapters with unique $A$ matrices are already discriminative, but normalizing does sharpen the distribution. The biased norms in the shared case completely prevent accurate discrimination, with the dbpedia adapter producing the largest average norm regardless of the dataset. Z-scoring significantly improves the results, leading to similar relative averages when compared to the traditionally trained LoRAs with unique $A$ matrices. The estimated norms could change post-calibration due to noisy estimation or distribution shift. We measure the sensitivity of performance with respect to these changes in Appendix E and find SEQR is robust to relatively large changes in the expected norms.

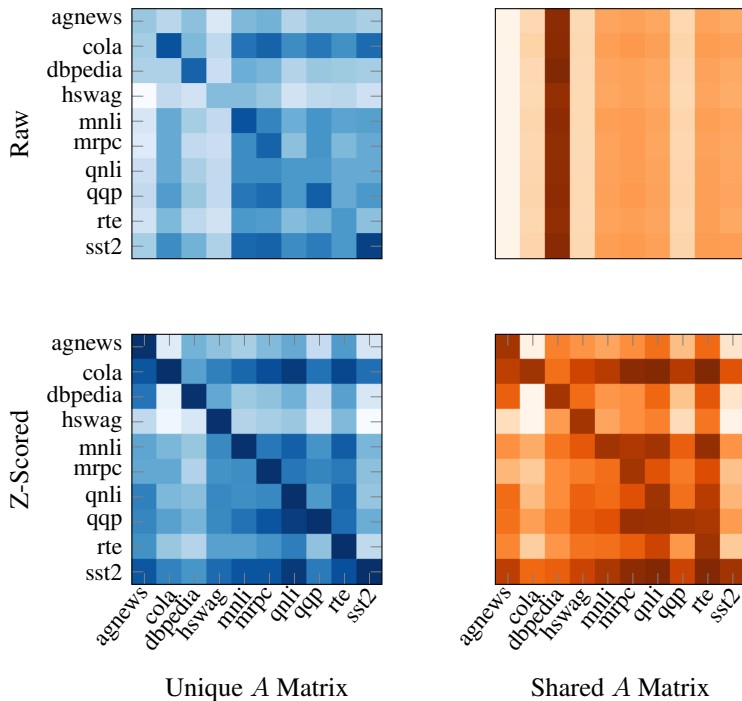

Figure 3: Raw (top) versus z-scored (bottom) activation norms for the adapters using unique (left) or shared (right) $A$ matrices. Rows are datasets and columns are the applied adapter.

## 4.4 ROUTING ACCURACY

We validate our theoretical results by measuring the percentage of tokens routed to the norm-maximizing adapter (Figure 4). ARROW chooses the adapter with the top singular vector most aligned to the input. This adapter is the norm-maximizing adapter for only 16% of tokens. The routing accuracy of LAG scales almost linearly with $k$, as LAG is equivalent to ARROW at $k = 1$ and equivalent to SPECTR at $k = 10$. SPECTR and SEQR both choose the norm-maximizing adapter in all cases. These results empirically confirm our theoretical findings and are consistent with prior work showing ARROW routing accuracies just above random change and improved routing with SPECTR (Fleshman & Van Durme, 2025b).

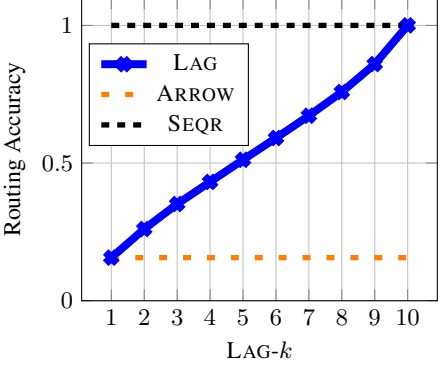

Figure 4: Routing accuracy as $k$ increases for LAG. LAG is equivalent to ARROW at $k = 1$ and to SPECTR at $k = 10$. ARROW chooses the norm-maximizing adapter for 16% of tokens.

## 4.5 TASK PERFORMANCE

We measure multi-task performance by evaluating the routing methods on the withheld data from each dataset. Keeping with previous work, we include MU-routing as an additional baseline (Ostapenko et al., 2024; Fleshman & Van Durme, 2025b). MU forgoes routing to individual LoRAs and instead computes the mean update using all adapters: $\mathbf{y} = W\mathbf{x} + \frac{1}{N}\sum_{i=1}^{N} B_i A\mathbf{x}$. While simple, averaging adapters can lead to poor performance due to interference in parameter space, especially with a large number of adapters (Ortiz-Jimenez et al., 2023; Tang et al., 2024; Stoica et al., 2025).

Fleshman & Van Durme (2025a) use $k = 20$ with LAG, filtering their adapter library to 2% of the total before using SPECTR to make the final selection. With only 10 adapters, we use a 30% reduction with $k = 3$ for demonstration purposes, but note the LAG task performance is equivalent to ARROW

Table 3: Mean and standard deviation of performance achieved across datasets and routing methods. SPECTR achieves identical performance as SEQR but at a higher computational cost.

| | | agnw | cola | dbpd | hswg | mnli | mrpc | qnli | qqp | rte | sst2 | AVG |
|---|---|---|---|---|---|---|---|---|---|---|---|---|
| Qwen-1.5B | MU | 77 ±0.3 | 93 ±0.3 | 94 ±0.1 | 52 ±0.7 | 73 ±0.0 | 85 ±0.1 | 94 ±0.3 | 95 ±0.4 | 93 ±0.1 | 93 ±0.5 | 84.8 |
| | ARW | 98 ±0.7 | **97** ±0.6 | **99** ±0.0 | 94 ±0.5 | **77** ±2.2 | 93 ±1.2 | 96 ±1.3 | 98 ±1.0 | 97 ±1.2 | 98 ±1.5 | 94.8 |
| | LAG | 99 ±0.3 | 96 ±2.3 | **99** ±0.8 | 93 ±0.7 | 75 ±0.7 | 93 ±0.4 | **97** ±0.1 | 98 ±1.1 | 98 ±0.7 | **99** ±1.1 | 94.8 |
| | SQR | **100** ±0.3 | 96 ±1.8 | **99** ±1.0 | **95** ±0.9 | 75 ±0.1 | **94** ±1.0 | **97** ±0.5 | **99** ±0.7 | **99** ±0.4 | **99** ±0.3 | **95.1** |
| Llama-3B | MU | 16 ±9.7 | 86 ±3.6 | 89 ±2.6 | 53 ±0.0 | 48 ±2.5 | 76 ±4.5 | 79 ±0.6 | 61 ±4.4 | 73 ±1.1 | 94 ±0.3 | 67.5 |
| | ARW | 89 ±0.9 | 92 ±1.6 | **100** ±0.1 | 78 ±12.2 | 78 ±2.1 | 92 ±1.2 | 92 ±2.8 | **97** ±0.9 | 94 ±3.1 | 97 ±1.0 | 90.9 |
| | LAG | 89 ±1.0 | 94 ±0.8 | **100** ±0.2 | 86 ±2.1 | **81** ±6.2 | **93** ±1.5 | 95 ±0.9 | **97** ±0.2 | **96** ±1.5 | **98** ±0.9 | 92.9 |
| | SQR | **91** ±0.5 | **96** ±0.9 | **100** ±0.2 | **87** ±2.3 | **81** ±7.5 | **93** ±2.2 | **96** ±1.4 | **97** ±0.1 | **96** ±2.1 | **98** ±2.3 | **93.5** |
| Qwen-7B | MU | 91 ±0.2 | 95 ±0.1 | 99 ±0.0 | 88 ±0.3 | 77 ±0.4 | 69 ±0.8 | 73 ±0.6 | 91 ±0.5 | 90 ±0.3 | 95 ±0.2 | 86.8 |
| | ARW | **100** ±0.2 | 98 ±1.2 | **100** ±0.2 | **100** ±0.5 | 97 ±0.5 | 95 ±1.4 | 96 ±0.4 | **99** ±0.1 | **98** ±0.1 | **99** ±0.4 | 98.1 |
| | LAG | **100** ±0.3 | 98 ±0.8 | **100** ±0.1 | **100** ±0.5 | 97 ±0.4 | 96 ±1.2 | **97** ±0.4 | **99** ±0.2 | 97 ±0.5 | **99** ±0.3 | 98.5 |
| | SQR | **100** ±0.1 | **99** ±0.6 | **100** ±0.1 | **100** ±0.6 | **98** ±0.8 | **98** ±0.9 | **97** ±0.8 | **99** ±0.3 | 97 ±0.3 | **99** ±0.2 | **98.7** |
| Llama-8B | MU | 0 ±0.1 | 88 ±8.9 | 34 ±13.9 | 66 ±0.7 | 66 ±0.8 | 84 ±0.8 | 83 ±3.7 | 64 ±7.4 | 70 ±20.5 | 49 ±27.8 | 60.4 |
| | ARW | 98 ±0.4 | 96 ±0.4 | **99** ±0.2 | 98 ±0.7 | **92** ±1.8 | 94 ±1.2 | 97 ±1.8 | **98** ±0.4 | 95 ±0.1 | **98** ±0.2 | 96.5 |
| | LAG | **99** ±0.4 | 97 ±0.8 | 98 ±0.7 | **99** ±0.6 | 91 ±0.9 | **95** ±1.3 | 97 ±1.6 | **98** ±0.4 | 97 ±0.6 | **98** ±0.5 | 96.8 |
| | SQR | **99** ±0.6 | **98** ±0.5 | **99** ±0.6 | **99** ±1.0 | 91 ±0.5 | **95** ±1.2 | **98** ±0.7 | **98** ±0.3 | **98** ±0.8 | **98** ±0.6 | **97.2** |

for $k = 1$ and to SPECTR and SEQR at $k = 10$. We control for variation in task difficulty by dividing each score by the performance of the correct adapter (Ostapenko et al., 2024; Fleshman & Van Durme, 2025b). This normalization captures the percentage of the upper bound performance achieved by routing with a perfect oracle. The oracle scores are included in Appendix G. We report the mean performance and standard deviation over three random seeds in Table 3. SEQR and SPECTR route equivalently, so we only include SEQR in the table. SEQR achieves the highest average score in all cases, outperforming MU, ARROW, and LAG.[5] The similar task-performance with LAG and identical performance with SPECTR make differences in efficiency a primary consideration for choosing among the various approaches. Next, we explore these differences in more detail.

## 4.6 ROUTING EFFICIENCY

SEQR yields the same improved multi-task performance as SPECTR, but with far greater efficiency. We measure the realized FLOPs and peak GPU memory used by each approach under various conditions (Figure 5). Total memory usage is dominated by the storage of the adapter library, so SEQR and ARROW are around twice as efficient when using shared $A$ matrices. SPECTR and LAG require storing unique $\hat{A}_i$ matrices per adapter, even when the original $A$ matrix is shared. SEQR stores an extra $Nr^2$ parameters for the $R_i$ matrices while ARROW stores an extra $Nn$ for the arrow vectors. This gives SEQR an additional advantage in storage costs when $r^2 < n$. For computation, SEQR provides a significant reduction in FLOPs over other methods, especially for large adapter libraries using a smaller LoRA rank per adapter. ARROW requires fewer FLOPs than SEQR when

---

[5] A paired t-test produces $p = 3.78e^{-4}$ when comparing with ARROW and $p = 3.7e^{-5}$ with LAG.

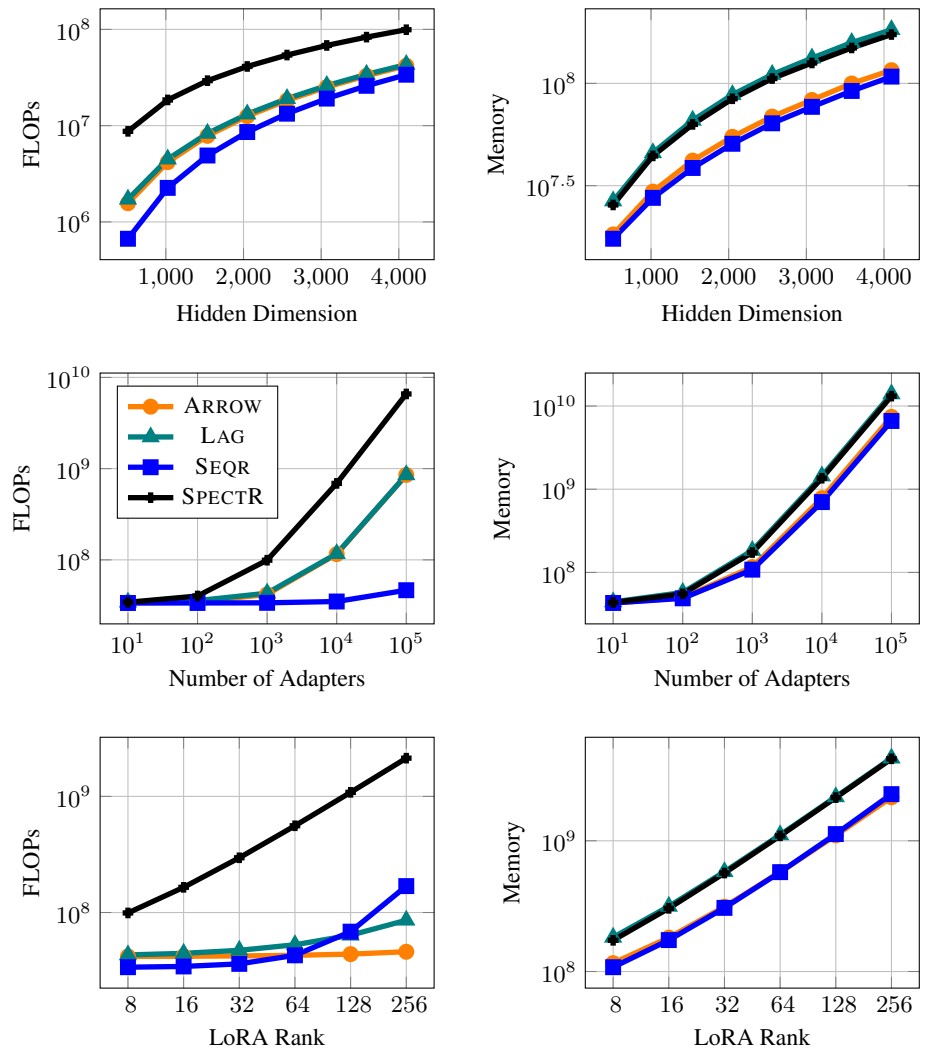

Figure 5: FLOPs (left) and GPU bytes used (right) for each method while varying hidden dimension (top), number of adapters in library (middle), and LoRA rank (bottom). Settings are fixed to $n = 4096$, $N = 1,000$, and $r = 8$ when not under evaluation. LAG uses $k = 20$ for ARROW filtering.

$r > \sqrt{n}$, but the relative task-performance of ARROW degrades at higher rank, where routing decisions are still limited by the rank-1 prototypes (Fleshman & Van Durme, 2025b).

## 5 CONCLUSION

In conclusion, we introduced SEQR, a state-of-the-art unsupervised LoRA routing algorithm. We formalized the goal of unsupervised LoRA routing in terms of activation norm-maximization and provided theoretical results for previous routing methods under this framework. The approaches that guarantee selecting the norm-maximizing adapter had better multi-task performance in our experiments. We showed that SEQR has this guarantee while being orders of magnitude more efficient than existing alternatives. SEQR leverages prior work showing that similar performance can be achieved when using LoRAs with frozen $A$ matrices shared across adapters, a finding we empirically validate. Sharing the $A$ matrices resulted in a higher variance in activation norms, which we corrected via an offline calibration step. Calibration improved performance for SPECTR and SEQR, with SEQR being significantly faster in execution due to the increased efficiency. SEQR maintains the security benefits of other unsupervised methods, preventing data leakage without access to the LoRA weights.

## 6 REPRODUCIBILITY

We take several steps to ensure the reproducibility of our work. First, we provide our source code containing the functionality for producing our exact training datasets, random seeds, adapter training, and evaluation. We provide similar details in Section 4.1 and Appendix D. Our theoretical results contain step-by-step proofs in Appendix A, Appendix B and Appendix C. Finally, we provide a concise version of our new routing procedure in Appendix F.

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

## A    Arrow Proof

*Proof.* We will construct 2x2 LoRA adapters $C$ and $D$ and an input $\mathbf{x}$ that satisfy the condition of the theorem.

1. Define $C = B_1 A_1$:

   Let matrix $C = \begin{pmatrix} 2 & 0 \\ 0 & 1 \end{pmatrix}$.

   The singular values are $\sigma_1(C) = 2$ and $\sigma_2(C) = 1$. The right singular vector corresponding to $\sigma_1$ is $\mathbf{v}_C = \begin{pmatrix} 1 \\ 0 \end{pmatrix}$.

2. Define $D = B_2 A_2$:

   We construct $D$ from the singular value decomposition $D = USV^T$ and choose the components to satisfy the theorem.

   Let $U = I$ the identity.

Let the singular values be $\sigma_1(D) = 3$ and $\sigma_2(D) = 1$.

Let the right singular vectors be $\mathbf{v}_D = \frac{1}{\sqrt{2}}\begin{pmatrix} 1 \\ 1 \end{pmatrix}$ and $\frac{1}{\sqrt{2}}\begin{pmatrix} 1 \\ -1 \end{pmatrix}$.

$$D = USV^T = \begin{pmatrix} 1 & 0 \\ 0 & 1 \end{pmatrix}\begin{pmatrix} 3 & 0 \\ 0 & 1 \end{pmatrix}\begin{pmatrix} \frac{1}{\sqrt{2}} & \frac{1}{\sqrt{2}} \\ \frac{1}{\sqrt{2}} & -\frac{1}{\sqrt{2}} \end{pmatrix}^T = \begin{pmatrix} \frac{3}{\sqrt{2}} & \frac{3}{\sqrt{2}} \\ \frac{1}{\sqrt{2}} & -\frac{1}{\sqrt{2}} \end{pmatrix}.$$

3. Choose vector $\mathbf{x}$:

Let $\mathbf{x} = \mathbf{v}_C = \begin{pmatrix} 1 \\ 0 \end{pmatrix}$.

4. Verify inequality:

$$\begin{aligned} LHS &= \operatorname{argmax}_i |\mathbf{v}_i^T \mathbf{x}| \\ &= \operatorname{argmax}_i \{|\mathbf{v}_C^T \mathbf{x}|, |\mathbf{v}_D^T \mathbf{x}|\} \\ &= \operatorname{argmax}_i \{1, \frac{1}{\sqrt{2}}\} = \text{(adapter 1)}. \end{aligned}$$

$$\begin{aligned} RHS &= \operatorname{argmax}_i ||B_i A_i \mathbf{x}||_2 \\ &= \operatorname{argmax}_i \{||C\mathbf{x}||_2, ||D\mathbf{x}||_2\} \\ &= \operatorname{argmax}_i \{\left\|\begin{pmatrix} 2 \\ 0 \end{pmatrix}\right\|_2, \left\|\begin{pmatrix} \frac{3}{\sqrt{2}} \\ \frac{1}{\sqrt{2}} \end{pmatrix}\right\|_2\} \\ &= \operatorname{argmax}_i \{2, \sqrt{5}\} = \text{(adapter 2)}. \end{aligned}$$

$LHS \neq RHS.$

$\square$

## B SPECTR PROOF

*Proof.* Let $B \in \mathbb{R}^{m \times r}$, $A \in \mathbb{R}^{r \times n}$, and $\mathbf{x} \in \mathbb{R}^n$. So,

$$\begin{aligned} ||BA\mathbf{x}||_2 &= ||USV^T\mathbf{x}||_2 && \text{(Equation 1)} \\ &= ||U\hat{A}\mathbf{x}||_2 && \text{(Equation 3)} \\ &= \sqrt{||U\hat{A}\mathbf{x}||_2^2} \\ &= \sqrt{\mathbf{x}^T \hat{A}^T U^T U \hat{A}\mathbf{x}} && \text{(Definition of squared 2-norm)} \\ &= \sqrt{\mathbf{x}^T \hat{A}^T \hat{A}\mathbf{x}} && \text{(Orthonormal columns} \implies U^T U = I) \\ &= \sqrt{||\hat{A}\mathbf{x}||_2^2} \\ &= ||\hat{A}\mathbf{x}||_2 \end{aligned}$$

$\square$

## C  SEQR PROOF

*Proof.* Let $B \in \mathbb{R}^{m \times r}$, $A \in \mathbb{R}^{r \times n}$, $\mathbf{x} \in \mathbb{R}^n$, and $B = QR$ from Equation 4. So,

$$
\begin{aligned}
||BA\mathbf{x}||_2 &= ||QRA\mathbf{x}||_2 && \text{(substitution)} \\
&= \sqrt{||QRA\mathbf{x}||_2^2} \\
&= \sqrt{\mathbf{x}^T A^T R^T Q^T QRA\mathbf{x}} \\
&= \sqrt{\mathbf{x}^T A^T R^T RA\mathbf{x}} && \text{(Orthonormal columns} \implies Q^T Q = I) \\
&= \sqrt{||RA\mathbf{x}||_2^2} \\
&= ||RA\mathbf{x}||_2
\end{aligned}
$$

$\square$

## D  ADAPTER DETAILS

We fit LoRA adapters targeting all attention layers in the network (query, key, value, and output projection layers). We choose initial settings for the LoRAs using the `unsloth` hyperparameter guide.[6] We use rank-32 adapters with a LoRA $\alpha = 64$ and dropout of 0.05. We train for two epochs using a cosine schedule with warm-up ratio of 5% and a batch size of 8. We sweep learning rates in the set {5e-6, 1e-5, 2e-5, 5e-5, 1e-4, 2e-4, 5e-4, 1e-3, 2e-3, 5e-3} for each dataset, but share learning rates across random seeds. We provide our source code for ease of replication.

## E  NORM ROBUSTNESS

SEQR scores use the estimated mean and standard deviation of the activation norms seen in the training data. We measure how sensitive the performance of SEQR is with respect to these estimates. We evaluate over our entire testing dataset, adding noise to the estimated means before z-scoring. The noise simulates post-calibration distribution shift or a noisy estimation of the mean activations during calibration. We add gaussian noise with varying intensities to alter the means by up to a full standard deviation of the activation distribution. Figure 6 displays the average performance of SEQR as noise is added. SEQR is robust to estimation errors, and falls below the performance of LAG and ARROW only when the estimated statistics are significantly off (20% and 100% of $\sigma$ respectively).

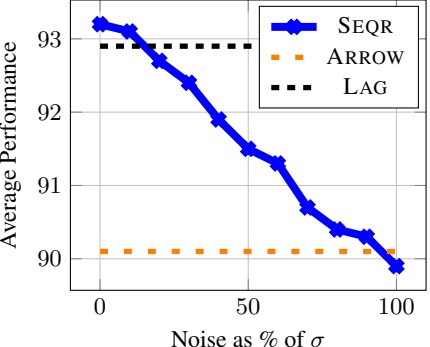

Figure 6: Average performance of SEQR as noise is added to the calibrated activation norm means with varying intensities.

---

[6] https://docs.unsloth.ai/get-started/fine-tuning-llms-guide/lora-hyperparameters-guide

# F SEQR ALGORITHM

We present the complete SEQR procedure in Algorithm 1.

---

**Algorithm 1** Secure and Efficient QR (SEQR) Routing

---

**Require:** Pretrained weight matrix $W \in \mathbb{R}^{m \times n}$
    Shared adapter matrix $A \in \mathbb{R}^{r \times n}$             $\triangleright$ Randomly initialized and frozen during training
    LoRA matrices $\{B_i \in \mathbb{R}^{m \times r}\}_{i=1}^{N}$
    Norm statistics $\{\mu_i, \sigma_i\}_{i=1}^{N}$                $\triangleright$ Estimated using training data

    **Preprocessing**
    **for** each adapter $B_i$ **do**
        Compute reduced QR decomposition: $B_i = Q_i R_i$          $\triangleright$ $B_i$ can be discarded
    **end for**

    **Inference** (given input $\mathbf{x} \in \mathbb{R}^n$)
    Compute shared intermediate representation: $\mathbf{z} \leftarrow A\mathbf{x}$
    **for** each adapter $i = 1, \ldots, N$ **do**
        Projected activation: $\mathbf{h}_i \leftarrow R_i \mathbf{z}$
        Score: $s_i \leftarrow (\|\mathbf{h}_i\|_2 - \mu_i)/\sigma_i$          $\triangleright$ Z-scored activation norm
    **end for**
    Select top adapter: $i^* \leftarrow \arg\max_i s_i$          $\triangleright$ Adapter with max activation norm
    Compute final output: $\mathbf{y} \leftarrow W\mathbf{x} + Q_{i^*}\mathbf{h}_{i^*}$          $\triangleright$ $Q_{i^*}\mathbf{h}_{i^*} = B_{i^*}A\mathbf{x}$
    **return y**

---

# G ORACLE PERFORMANCE

We include the raw task performance of each model when using the ground truth adapter. This simulates perfect routing, an upper bound on expected performance. All scores in Section 4 are normalized by these values to control for differences in task difficulty or model capability.

Table 4: Performance using perfect ground truth routing.

|  | agnews | cola | dbped | hswag | mnli | mrpc | qnli | qqp | rte | sst2 | AVG |
|---|---|---|---|---|---|---|---|---|---|---|---|
| Qwen-1.5B | 90.3 | 81.1 | 98.8 | 80.7 | 87.0 | 85.3 | 83.4 | 86.3 | 86.1 | 91.7 | 87.1 |
| Llama-3B | 90.0 | 78.9 | 99.0 | 81.5 | 85.7 | 85.0 | 85.5 | 86.3 | 87.9 | 92.8 | 87.3 |
| Qwen-7B | 91.2 | 85.1 | 98.8 | 90.5 | 90.6 | 86.7 | 88.5 | 87.5 | 92.5 | 93.9 | 90.5 |
| Llama-8B | 91.3 | 82.9 | 98.7 | 86.9 | 88.0 | 86.1 | 88.3 | 88.4 | 91.6 | 94.0 | 89.6 |

