# OpenReview forum: "SEQR: Secure and Efficient QR-based LoRA Routing"
_ICLR.cc/2026/Conference — Submitted to ICLR 2026_

### Official Review · Reviewer_jhfQ · 2025-10-25

**Soundness:** 2
**Presentation:** 3
**Contribution:** 3
**Rating:** 6
**Confidence:** 3

**Summary:**

This paper introduces SEQR, an unsupervised LoRA routing method designed for efficiency and security. The authors formalize the goal of unsupervised routing as activation norm-maximization, providing a theoretical framework to analyze existing methods and their proposed solution. SEQR leverages a shared, frozen A matrix and QR decomposition to provably select the norm-maximizing adapter with significantly lower computational cost. The claims are supported by theoretical proofs and empirical experiments demonstrating competitive multi-task performance and superior efficiency.

**Strengths:**

1. The paper's formalization of unsupervised routing as an activation norm-maximization problem provides a clear and valuable framework. The subsequent theoretical analysis, including proofs that SPECTR and SEQR are norm-maximizing while ARROW is not, adds significant rigor and clarifies the behaviors of these methods.

2. SEQR demonstrates a significant improvement in computational and storage efficiency over existing methods that offer similar routing guarantees, as shown in the complexity analysis and empirical measurements. This makes the method highly practical for real-world scenarios involving large libraries of LoRA adapters, directly addressing a key challenge in the field.

**Weaknesses:**

1. The entire SEQR framework is built upon the assumption that a single, frozen, randomly initialized A matrix is sufficient for training diverse adapters. While the paper shows this holds for the tested tasks, this constraint might limit performance when fine-tuning for highly dissimilar or complex tasks where a unique A matrix could capture important task-specific input transformations.

2. The paper posits activation norm-maximization as the primary goal for unsupervised routing. While intuitive and empirically supported, this objective may not be universally optimal. It is possible for an adapter to produce the highest activation norm without yielding the best task-specific performance, especially in cases of model miscalibration or overfitting.

3. SEQR requires an offline calibration step that computes and stores the mean (µ_i) and standard deviation (σ_i) of activation norms from each adapter's training data. While this avoids training a cross-silo router, these statistics themselves are derivatives of the private data and could potentially leak information about the data distribution.

4. According to Table 3 and footnote 5, the average performance improvement of SEQR over LAG (k=3) is small (93.5 vs 92.9), and the difference is not statistically significant (p=0.096). This suggests that SEQR's primary advantage is its efficiency, not necessarily superior task performance over a well-configured LAG.

5. The paper compares SEQR (which requires a shared A) against baselines like ARROW and LAG. It is unclear if these baselines were evaluated using adapters with unique A matrices (their standard formulation) or shared A matrices in the final performance table (Table 3), which impacts the fairness of the storage and computational comparisons.

**Questions:**

Please see weaknesses.

**Details Of Ethics Concerns:**

Null.

---

> ### Author Response · Authors · 2025-11-14
>
> Thank you for the review and unique points. We have submitted a revised version of the manuscript to address the concerns raised by all reviewers. The paper is much stronger with the changes, so we thank you again.
>
> Here is a summary of the revisions:
>
> - (Section 4.5) We expanded our evaluation to four models (Qwen 1.5B, Llama 3B, Qwen 7B, and Llama 8B) to demonstrate the effectiveness of the approach across different model families and sizes.
>
> - (Lines 322-323 & Appendix E) We test the robustness of calibration to estimation noise or distribution shift post-calibration and find that SEQR is robust to relatively large changes.
>
> - (Lines 92-99) We expand the related work to reference more LoRA variants and combination methods.
>
> - (Lines 411-412 & Appendix G) We are more explicit in providing an upper bound on performance using a perfect router, we include those scores in the Appendix.
>
> - (Appendix F) We move Algorithm 1 to the appendix to make room for the above changes.
>
> Specifically for your stated weaknesses:
>
> 1. We cite several works that show a single frozen A matrix is sufficient for learning across diverse tasks. Specifically, Zhu et al.’s Asymmetry in Low-Rank Adapters of Foundation Models, 2024., provides analysis on why “a random untrained A should perform nearly as well as a fine-tuned one.” Gaussian random projections have long been used as general feature extractors, and this is what a frozen A matrix in LoRA is doing. The B matrix can then learn over that dimensionality reduced space. The diversity of the tasks doesn’t matter as much since the A matrix is chosen independent of the data distribution. Then when it comes to routing, previous work has shown that diverse tasks are actually easier to route accurately versus similar tasks, which can make routing decisions more difficult due to the similarity of the data distributions (the cited SpectR paper explores this).
>
> 2. We establish norm-maximization as the goal based on the motivations of previous approaches (Arrow Routing and SpectR), which implicitly relate to norm-maximization without explicitly establishing that as the objective. We agree it is an intuitive goal and has been shown effective in other in- and out-of-distribution detection work as mentioned in the paper. No objective would be universally optimal, but SEQR (which targets this goal) is roughly 95% as accurate as a perfect router (Appendix G) in our experiments.
>
> 3. These statistics are adapter specific and are gathered independently across LoRAs. There isn’t any additional security concern when treated as 2 additional parameters for each LoRA.
>
> 4. We agree that the primary advantage of SEQR is efficiency (SpectR would have identical task performance by design). However, by expanding our experiments to 3 additional models, the statistical significance of SEQR over Arrow and LAG has increased. We updated the p-values, which now indicate there is enough evidence to support SEQR performing significantly better than the other approaches.
>
> 5. We ensure the comparisons are fair by taking advantage of shared A matrices if they benefit the other methods. Specifically, Arrow routing can benefit from the shared A matrices by reducing the storage/memory requirements. We call this out on Lines 235 and 425 and that is what is shown in Figure 5. SpectR and LAG don’t benefit from having a shared A since the SVD is used to recompute new As and Bs which would be unique per adapter.
>
> Thank you again for the constructive review, we hope the revision and discussion satisfies your concerns.

---

### Official Review · Reviewer_BjAp · 2025-10-30

**Soundness:** 3
**Presentation:** 3
**Contribution:** 3
**Rating:** 6
**Confidence:** 5

**Summary:**

In secure environments, supervised router training may lead to privacy leakage. To address this, we propose SEQR, an efficient unsupervised LoRA routing algorithm. Theoretically, SEQR can accurately identify the norm-maximizing adapter with high efficiency, making it a scalable and effective solution for dynamic LoRA composition. Extensive experiments verify the rationality and effectiveness of the proposed method.

**Strengths:**

This paper is theoretically sound, well-motivated, and proposes a simple yet effective method.

**Weaknesses:**

Although the theoretical motivation of this method is reasonable, several questions remain:

1. While this is an unsupervised LoRA routing algorithm, supervised routing methods might serve as an upper performance bound — it would be helpful to include such results for reference.
2. The datasets used in the experiments appear somewhat outdated; testing on more recent datasets commonly used for LLM evaluation and comparing with results from LoRA-only fine-tuning would strengthen the work.
3. It is recommended to provide results on a wider range of large language models (LLMs) to further demonstrate the generality of the proposed method.

**Questions:**

This is not a critical issue to address, and adding new experiments might be difficult during the rebuttal stage. However, providing relevant results would make the work more convincing.

---

> ### Author Response · Authors · 2025-11-14
>
> We appreciate your review and the opportunity to improve our paper. We have submitted a revised version of the manuscript to address the concerns raised by all reviewers. The paper is much stronger with the changes, so we thank you again.
>
> Here is a summary of the revisions:
>
> - (Section 4.5) We expanded our evaluation to four models (Qwen 1.5B, Llama 3B, Qwen 7B, and Llama 8B) to demonstrate the effectiveness of the approach across different model families and sizes (addresses weakness 3).
>
> - (Lines 322-323 & Appendix E) We test the robustness of calibration to estimation noise or distribution shift post-calibration and find that SEQR is robust to relatively large changes (addresses comment by another reviewer).
>
> - (Lines 92-99) We expand the related work to reference more LoRA variants and combination methods (addresses comment by another reviewer).
>
> - (Lines 411-412 & Appendix G) We are more explicit in providing an upper bound on performance using a perfect router, we include those scores in the Appendix (addresses weakness 1).
>
> - (Appendix F) We move Algorithm 1 to the appendix to make room for the above changes.
>
> We added the oracle routing scores in Appendix G to provide the upper bound on performance that you requested. These are the scores achieved by a perfect router. The scores in table 3 are the percentage of that upper bound achieved by each method. On average, SEQR is over 93-99% as effective as the perfect router in our experiments.
>
> We use the same datasets as previous work in this area to make comparisons easier. We’d argue that these datasets are still frequently used, and many newer datasets also include examples from these in a mixture. As we see in Appendix G, LoRA only fine-tuning still only achieves around 80-90% accuracy across the four models we tested. Our theoretical results are dataset agnostic, SEQR will make the same routing decisions as SpectR regardless of dataset while being far more efficient.
>
> Thank you again for the thoughtful review, we hope the revision and discussion satisfies any concerns.

---

### Official Review · Reviewer_AiX7 · 2025-10-31

**Soundness:** 3
**Presentation:** 3
**Contribution:** 3
**Rating:** 4
**Confidence:** 5

**Summary:**

This paper formalizes unsupervised LoRA routing as an activation norm maximization problem and proves that SPECTR is equivalent to directly maximizing this norm. Under the setting of a shared and frozen A, the paper proposes SEQR, which applies QR decomposition and spatial comparison to each B_i. This approach maintains the strict optimal routing guarantee while reducing computational complexity.

**Strengths:**

1.This paper explicitly defines unsupervised routing as an optimization problem of maximizing the activation norm within a LoRA library, providing a unified analytical baseline for different methods.
2.The proposed algorithm offers significant efficiency advantages, requiring only r*r multiplication for inference, thereby lowering algorithmic complexity. Experiments demonstrate that under typical scales, routing FLOPs are reduced from the million-level to 64K.

**Weaknesses:**

1.The equivalence and efficiency of SEQR depend on the assumption that "all adapters share the same frozen A matrix." Does the method become inapplicable in scenarios where A is trained independently for each task or subsequently fine-tuned?
2.The paper's experimental scope is limited, evaluating only on a relatively small base model (Llama-3.2-3B) and classification tasks. Evaluations on other types or scales of LLMs are missing.
3.The paper's motivation emphasizes security and privacy, but SEQR only associates this with being unsupervised and sharing matrix A. Its core algorithmic innovation (i.e., QR decomposition) does not intrinsically provide additional security or privacy guarantees, which does not seem to fully support this claim.

**Questions:**

-

---

> ### Author Response · Authors · 2025-11-14
>
> Thank you for your review and highlighting the strengths of our work. We have submitted a revised version of the manuscript to address the concerns raised by all reviewers. The paper is much stronger with the changes, so we thank all reviewers again.
>
> Here is a summary of the revisions:
>
> - (Section 4.5) We expanded our evaluation to four models (Qwen 1.5B, Llama 3B, Qwen 7B, and Llama 8B) to demonstrate the effectiveness of the approach across different model families and sizes (addresses weakness 2).
>
> - (Lines 322-323 & Appendix E) We test the robustness of calibration to estimation noise or distribution shift post-calibration and find that SEQR is robust to relatively large changes (addresses comment by other reviewer).
>
> - (Lines 92-99) We expand the related work to reference more LoRA variants and combination methods (addresses comment by other reviewer).
>
> - (Lines 411-412 & Appendix G) We are more explicit in providing an upper bound on performance using a perfect router, we include those scores in the Appendix (addresses comment by other reviewer).
>
> - (Appendix F) We move Algorithm 1 to the appendix to make room for the above changes.
>
> Your weakness/question #1 asks about the applicability of SEQR to scenarios where the A matrices are independently fine-tuned. We contribute the theoretical results involving Arrow, LAG, and SpectR in that scenario, but SEQR is designed specifically to take advantage of shared A matrices. That said, it is possible to align the A matrices of LoRAs trained independently with techniques such as (Stoica et al., Model Merging with SVD to Tie the Knots, 2024.). This would allow the use of SEQR in those scenarios, but would break the security aspect we are focused on in this work.
>
> For stated weakness #3, could you clarify which claim is not fully supported? The core innovation of SEQR is the combination of using a frozen and shared A matrix along with the QR decomposition of the B matrices. Keeping the matrix frozen maintains the security of unsupervised approaches, and sharing A enables efficient routing via the QR decomposition. Our other contributions include the analysis of other approaches for secure routing.
>
> Thank you again for taking the time to review our paper, we hope we have satisfied your concerns.

---

### Official Review · Reviewer_8k6Y · 2025-11-07

**Soundness:** 3
**Presentation:** 3
**Contribution:** 2
**Rating:** 4
**Confidence:** 4

**Summary:**

The paper proposes SEQR, an unsupervised routing algorithm for selecting LoRA adapters based on activation norm maximization. The authors formalize unsupervised LoRA routing as the problem of efficiently identifying the adapter that maximizes the activation norm. SEQR targets the shared-A setting and precomputes a reduced QR decomposition, achieving the same norm-maximization guarantee as SpectR but with substantially lower complexity. Experiments show the benefits of the proposed method.

**Strengths:**

1.	Clear problem formalization: Casting unsupervised LoRA routing as activation norm maximization provides a coherent lens to compare methods and reason about guarantees.
2.	The method is simple yet effective to follow.

**Weaknesses:**

1.	Incremental novelty: Both building blocks—SVD/QR factorization of LoRA variants and the Shared A regime—are established ideas in the LoRA literature and in modular LoRA routing. SEQR primarily combines “known” linear-algebraic decompositions with the shared-A assumption and applies them to norm-maximizing zero-shot routing. While the formalization is useful and the efficiency result is practical, the conceptual leap beyond these ingredients is modest.
2.	Narrow baselines and single-family backbone: This paper evaluates primarily on a single Llama-3.2-3B-Instruct model and compares with works from the same series. This limits the external validity and makes it hard to assess robustness across model families, sizes, and tasks beyond the included classification suite.
3.	Limited related work: The related work section lacks a broader overview of LoRA variants and composition methods, which would better situate SEQR within the ecosystem.
4.	Ablations are insufficient: The offline calibration step is central under shared-A but lacks robustness ablations (distribution shift, noisy μ/σ, per-layer vs aggregated statistics). The paper validates how hidden dimension, number of adapters, and LoRA rank affect routing efficiency, but does not show how these factors impact task performance.

**Questions:**

see weaknesses.

---

> ### Author Response · Authors · 2025-11-14
>
> Thank you for taking the time to leave a thoughtful and constructive review. We have submitted a revised version of the manuscript to address the concerns raised by all reviewers. The paper is much stronger with the changes, so we thank you again.
>
> Here is a summary of the revisions:
>
> - (Section 4.5) We expanded our evaluation to four models (Qwen 1.5B, Llama 3B, Qwen 7B, and Llama 8B) to demonstrate the effectiveness of the approach across different model families and sizes (addresses weakness 2).
>
> - (Lines 322-323 & Appendix E) We test the robustness of calibration to estimation noise or distribution shift post-calibration and find that SEQR is robust to relatively large changes (addresses weakness 4).
>
> - (Lines 92-99) We expand the related work to reference more LoRA variants and combination methods (addresses weakness 3).
>
> - (Lines 411-412 & Appendix G) We are more explicit in providing an upper bound on performance using a perfect router, we include those scores in the Appendix (addresses comment from another reviewer).
>
> - (Appendix F) We move Algorithm 1 to the appendix to make room for the above changes.
>
> We disagree with the stated weakness 1. QR decompositions have been used for LoRA initializations or to constrain updates, but we are unaware of any work that uses QR to inform unsupervised routing decisions. Similarly, shared A matrices have been established in the federated learning space or for parameter savings but using frozen and shared A matrices to make routing more efficient while maintaining security is unexplored. Novelty should be disentangled from complexity, as simpler solutions should always be preferred.
>
> Thank you again for the constructive review, we hope our revision satisfies your concerns.

---

### Meta-Review · Area_Chair_ioeG · 2026-01-02

**Summary:**

This paper formalizes unsupervised LoRA routing as an activation norm maximization problem and then proposes SEQR, an unsupervised routing algorithm for selecting LoRA adapters. The authors prove that SPECTR is equivalent to directly maximizing this norm. Under the setting of a shared and frozen A, the proposed SEQR applies QR decomposition and spatial comparison to each B_i. This approach maintains the strict optimal routing guarantee while reducing computational complexity.

I also read this paper by myself, and found that the motivation is still unclear. More importantly, the novelty based on SVD is not significant as this is well stuided for LoRA. Besides, the assumption that "all adapters share the same frozen A matrix" is too strong. I suggest the authors to justify this assumption and improve the performance. I didn't find a significant improvement over LAG.

**Reviewer Concerns:**

Reviewer 8k6Y concerns about the incremental novelty, narrow baselines, limited related work, and ablations. The later three points were addressed but the first one cannot.

Reviewer AiX7 concerns about the strong assumption, limited experiments, unclear motivation. The experiments can be conducted but the assumption and motivation were still not well stated during the rebuttal.

Reviewer BjAp concerns about the experiments.

Reviewer jhfQ concerns about the experiments, the assumption, the possibility of data distribution leak.

**Reviewer Scores:**

I think the review comments are clear and I'm not sure all of the reviewers will change their score.

---

### Decision · Program_Chairs · 2026-01-26

Reject